# Association of relative handgrip strength on the development of diabetes mellitus in elderly Koreans

Yeo Ju Sohn[1], Hong Soo Lee[2], Hasuk Bae[3], Hee Cheol Kang[4], Hyejin Chun[1], Insun Ryou[1], Eun Jee Chang[5], Sungchan Kang[5], Sang Wha Lee[2]*, Kyung Won Shim[1]*

1 Department of Family Medicine, Ewha Womans University Seoul Hospital, Ewha Womans University College of Medicine, Seoul, Korea, 2 Department of Family Medicine, Ewha Womans University Mokdong Hospital, Ewha Womans University College of Medicine, Seoul, Korea, 3 Department of Rehabilitation Medicine, Ewha Womans University Mokdong Hospital, Ewha Womans University College of Medicine, Seoul, Korea, 4 Department of Family Medicine, Severance Hospital, Yonsei University College of Medicine, Seoul, Korea, 5 Graduate School of Public Health, Seoul National University, Seoul, Korea

☯ These authors contributed equally to this work.
* ghwa@ewha.ac.kr (SWL); ewhashim@ewha.ac.kr (KWS)

**Data Availability Statement:** All data are in the manuscript and/or Supporting information files.

**Funding:** The author(s) received no specific funding for this work.

## Abstract

### Background

Diabetes mellitus (DM) is a significant public health concern, particularly in the elderly population. Absolute handgrip strength (HGS) serves to quantify muscle strength. It is recommended that the risk of low muscle strength and increased body mass index be concurrently evaluated using relative HGS. There are currently insufficient evidence regarding the relationship between relative HGS and DM in the elderly Korean population. Therefore, the association between relative HGS and the development of DM in Korean elderly was investigated.

### Methods

Data from the Korean Longitudinal Study of Ageing were used to determine the odds ratio (OR) between relative HGS and DM during the follow-up period from 2006–2020 among Korean men and women aged ≥65 years without DM when they first participated in this survey. Analysis was conducted using the Generalized Estimating Equation method. Trend analysis was performed for DM development based on relative HGS.

### Results

Among elderly males, higher relative HGS groups had reduced odds of developing DM (Middle tertile: OR 0.87, 95% CI 0.61–1.23, p = 0.419.) (Upper tertile: OR 0.82, 95% CI 0.56–1.18, p = 0.281.) Among elderly females, the reductions were similar. (Middle tertile: OR 0.82, 95% CI 0.66–1.03, p = 0.087.) (Upper tertile: OR 0.79, 95% CI 0.50–1.25, p = 0.306.) However, these differences were not statistically significant. Significant predictors of new-onset DM included age, BMI (overweight/obese), household income, alcohol consumption,

**Competing interests:** The authors have declared that no competing interests exist.

hypertension, and chronic liver disease. Trend tests indicated a substantial decrease in the OR as the relative HGS increased for male and total groups (p for trend < 0.05).

## Conclusion

Relative HGS did not achieve statistical significance. Our findings indicate that BMI, particularly overweight and obesity, significantly predicts new-onset DM. However, trend tests indicated a substantial decrease in the OR as the relative HGS increased for male and total groups (p for trend < 0.05), even after adjusting for BMI categories. Despite the lack of statistical significance in some cases, the trend suggests that promoting resistance exercises to enhance HGS could be beneficial in DM prevention. Comprehensive DM prevention strategies should include managing obesity and chronic conditions for elderly.

## Introduction

The global prevalence of diabetes mellitus (DM) is a growing concern, with an estimated 529 million cases in 2021, projected to reach 1.31 billion by 2050 [1]. Managing DM in older adults is particularly challenging due to insulin resistance, declining pancreatic function, and associated complications such as cardiovascular diseases and cerebrovascular diseases [2, 3]. Managing DM in older adults requires specialized care involving regular monitoring of blood glucose levels, dietary monitoring, exercise, medication, and consistent medical visits. This underscores the need for research and strategies to prevent the onset of DM in elderly populations.

Absolute handgrip strength (HGS) is a rapid, cost-effective, and noninvasive measure of muscle strength, a crucial health indicator in older adults [4]. Low HGS adversely affects independence, daily functioning, and quality of life in the elderly population. HGS is an indicator with a significant correlation with sarcopenia, which refers to muscle loss with age, and serves as a basic indicator to evaluate overall physical ability and muscle function, especially in the elderly population [5]. According to previous studies, it has been revealed that there is a correlation between BMI and HGS [6–8]. Previous studies have demonstrated the usefulness of absolute HGS in the identification of various health problems and its potential as a new vital sign across the lifespan, but have shown conflicting results [9]. The confounding effect of body size was thought to be one of the causes. In other words, previous studies conducted only with the concept of absolute HGS showed inconsistent results, and this was the result of using absolute HGS as an indicator of muscle strength without body mass correction, as absolute HGS is closely related to body mass index. Relative HGS has therefore been recommended as a better metric to take into account the effects of both body mass and muscle strength, and this measure helps account for differences in HGS that may be influenced by an individual's overall size. Therefore, in large-scale studies or clinical trials related to muscle strength, it is recommended to simultaneously evaluate the risk of increased body mass and low muscle strength using relative HGS, which is the value of absolute HGS divided by BMI and can act as a confounding variable [10]. Various previous studies have demonstrated that relative HGS may be beneficial in predicting cardiovascular biomarkers, metabolic profile, and risk of other cardiometabolic disorders [11–13].

Resistance strength training using light dumbbells during physical activity is recommended to prevent the development of DM in older adults with reduced muscle strength. Therefore, a decrease in the relative HGS, used as an indicator of muscle strength, may increase the risk of

developing DM [14]. In addition, there are previous research results showing that relative HGS can predict new-onset DM better than absolute HGS. A study conducted in middle-aged and older adults in Europe also found that low HGS was an independent predictor of new-onset DM risk, suggesting that relative HGS had a slightly higher predictive ability for future DM than absolute HGS in people aged 50 years or older. have emphasized that screening for low HGS may be of value in preventing DM [12]. However, previous studies show insufficient evidence for an association between relative HGS and DM in the elderly Korean population. Moreover, most of the previous studies that have investigated the relationship between relative HGS and DM in Koreans were cross-sectional studies that confirmed the relationship between relative HGS and DM simultaneously [15–17].

Therefore, this study aimed to investigate the association between relative HGS and the development of DM in Korean men and women aged ≥65 years, using data from the Korean Longitudinal Study of Ageing. (KLoSA). To further elucidate the impact of BMI on DM incidence, we incorporated trend analysis in our study. This approach allows us to assess the incremental effect of BMI categories (underweight, normal, overweight, and obese) on the risk of developing DM, providing a more nuanced understanding of this relationship.

## Materials and methods

### Data source and study sample

This study analyzed individuals aged ≥65 years using data from the 1st to 8th waves (2006–2020) of the KLoSA. The KLoSA, provided by the Korea Employment Information Service, is a public dataset that provides information on the social, economic, demographic, and health conditions of older adults. The approval number from Statistics Korea was 336002, and the approval date was March 30, 2006. The survey content included household background, personal attributes, family, health, employment, income, consumption, assets, subjective expectations, quality of life, and questions about deceased individuals. The survey also included computer-assisted, face-to-face interviews.

This study used the data collected by the KLoSA (2006–2020), involving repeated measurements of 33,701 elderly individuals aged ≥65 years. This dataset excluded 7,379 entries with missing values in the analysis of absolute HGS or BMI, resulting in 26,322 entries. Among these, 4,306 entries (1,246 individuals) with DM in the first survey or those undergoing DM treatment were excluded, resulting in a final sample of 22,016 participants who met the inclusion criteria. Fig 1 depicts a flowchart of the data selection process, and the initial participant counts for each survey year are highlighted in bold in Table 1.

### Ethical approval and informed consent to participates

The KLoSA was approved by the state government in accordance with Article 18 of the Statistics Act, and the National Statistical Office approval number was 336002 and the approval date was March 30, 2006. This survey was conducted after obtaining informed consent from research participants. The responses were kept confidential in accordance with Article 33 (Protection of Confidentiality) and Article 34 (Obligations of Statistical Workers) of the Statistics Act, and were not used for any purpose other than statistical purposes. Anonymized data can be used and downloaded by the public on the survey website (https://survey.keis.or.kr/klosa/klosa04.jsp). This study obtained permission to use public data, downloaded data from the KLoSA website, and performed secondary data analysis. Therefore, this study did not harm participants and anonymity and confidentiality were guaranteed. The database contains only de-identified data. Therefore, this study did not pose any risk to study subjects. Researchers for this study accessed KLoSA data on July 1, 2023, for research purposes.

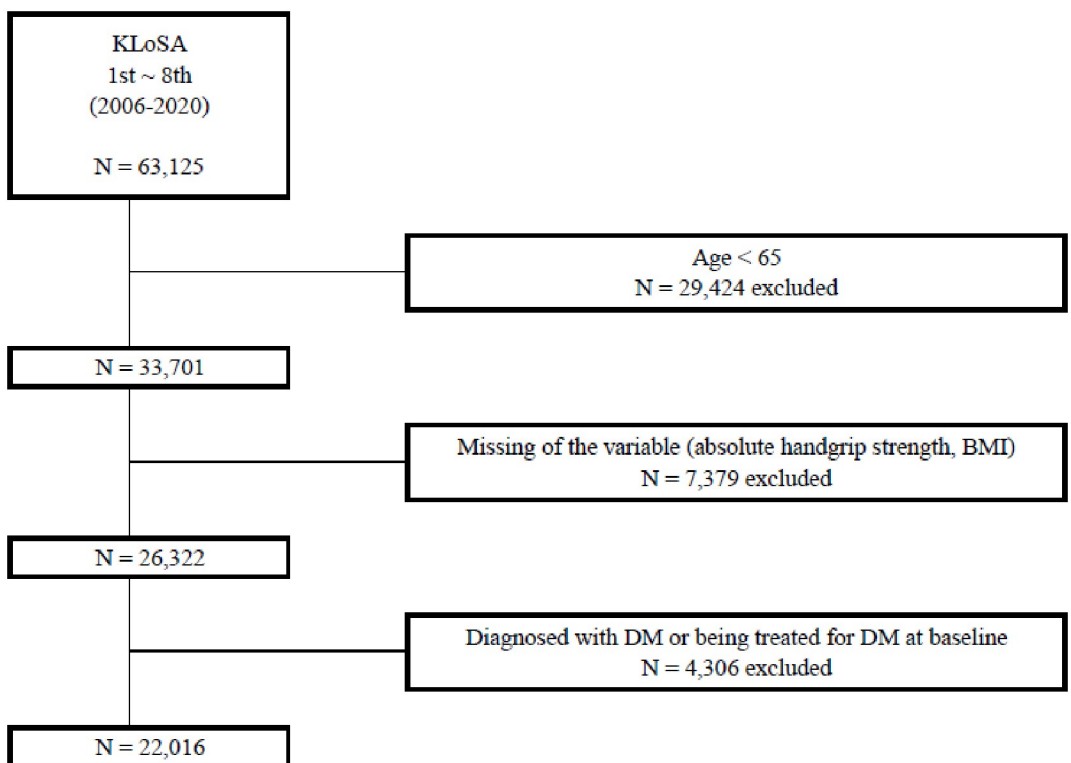

**Fig 1. Flow diagram of participants meeting the inclusion and exclusion criteria.** * KLoSA = Korean Longitudinal Study of Ageing.

## Dependent variable

The dependent variable in this study, referred to as the outcome variable, was the occurrence of DM between 2008 and 2020. This was defined as a positive response to the question, "Since the last survey, have you received a diagnosis of DM from a doctor, or have you been told that

**Table 1. Initial participants by wave\*.**

| First participation wave | Wave by order | | | | | | | | Total |
|---|---|---|---|---|---|---|---|---|---|
| | **1** | **2** | **3** | **4** | **5** | **6** | **7** | **8** | |
| 1 | **2,872** | 2,146 | 1,817 | 1,597 | 1,340 | 1,164 | 1,016 | 850 | 12,802 |
| 2 | 0 | **608** | 450 | 408 | 381 | 352 | 310 | 280 | 2,789 |
| 3 | 0 | 0 | **381** | 306 | 268 | 263 | 238 | 225 | 1,681 |
| 4 | 0 | 0 | 0 | **389** | 329 | 307 | 280 | 275 | 1,580 |
| 5 | 0 | 0 | 0 | 0 | **381** | 325 | 282 | 287 | 1,275 |
| 6 | 0 | 0 | 0 | 0 | 0 | **338** | 271 | 274 | 883 |
| 7 | 0 | 0 | 0 | 0 | 0 | 0 | **348** | 299 | 647 |
| 8 | 0 | 0 | 0 | 0 | 0 | 0 | 0 | **359** | 359 |
| Total | 2,872 | 2,754 | 2,648 | 2,700 | 2,699 | 2,749 | 2,745 | 2,849 | 22,016 |

\* wave = survey year number (Wave means distinct periods of time in which answers are collected from respondents.)

\* 1st wave = 2006, 2nd wave = 2008, 3rd wave = 2010, 4th wave = 2012, 5th wave = 2014, 6th wave = 2016, 7th wave = 2018, 8th wave = 2020 [year]

\* Initial participants are indicated in bold.

your blood sugar is high?" or a positive change from "no" to "yes" in response to the question, "Are you currently taking medication for DM or undergoing treatment to stabilize blood sugar levels?" Responses to these two questions were combined to create a variable for the occurrence of DM.

### Independent variable

The independent variable in this study was relative HGS, which was calculated as the ratio of absolute HGS to BMI. Relative HGS can be calculated as follows:

$$\text{Relative HGS } [m^2] = \frac{\text{Absolute HGS } [kg]}{\text{Body Mass Index } \left[\frac{kg}{m^2}\right]}$$

Absolute HGS was measured in kg, whereas relative HGS was in $m^2$. The HGS gauge used in the KLoSA was the Tanita 6103 model. HGS measurement was conducted with the principle of identifying whether the respondent is in a state where measurement is possible, and measurement is not conducted if the respondent does not want to or if one hand is currently injured or in pain. After confirming the possibility of HGS measurement, the HGS of the primarily used hand was measured first, followed by the measurement of the HGS of the other hand. The absolute HGS was determined by confirming the ability to measure HGS, identifying the dominant hand, and using an HGS meter to measure the HGS of both hands twice, with the average value representing the overall HGS. Previous studies analyzing HGS have shown differences in strength among the sexes owing to the physical differences. Therefore, this study conducted a stratified analysis by sex and converted the continuous variable of relative HGS into a categorical variable with three groups (High, Middle, and Low) based on the tertiles for each sex. We defined cutoffs for underweight (<18.5 kg/m2), normal (18.5–22.9 kg/m2), overweight (23.0–24.9 kg/m2), and obesity(≥25.0 kg/m2) based on the WHO Asia-Pacific regional guidelines [18].

### Control variables

Control variables were broadly categorized into sociodemographic, economic, and health-related behavior and chronic disease factors. Sociodemographic factors included sex, age, educational level, residential area, and marital status. Economic factors included household income. Health-related behavior and chronic disease factors included smoking status, alcohol consumption, regular physical activity, Activities of Daily Living (ADL) index, Instrumental Activities of Daily Living (IADL) index, and diagnosis of chronic diseases (hypertension, heart disease, cerebrovascular disease, cancer, chronic respiratory disease, and chronic liver disease). Participants were asked whether they had ever been diagnosed with the chronic diseases by a doctor and answered yes or no. Although BMI is not a control variable, analysis was performed by including it in the GEE model to evaluate collinearity.

### Analytical approach and statistics

This study used data from the first (2006) to the eighth (2020) panels conducted every two years as part of the KLoSA, to analyze the relationship between HGS and new-onset DM using the generalized estimating equation (GEE) model. As panel data involve multiple observations for the same individual at different time points, making yearly observations is often not independent; therefore, GEE model was used to account for within-subject correlations [19–21]. Since correlations exist between measurements in medical research, GEE should be applied during data analysis [22]. In the GEE, an analysis was performed to estimate parameters using

a working correlation matrix to explain the correlation between dependent variables. Quasi-likelihood information criterion (QIC) and quasi-likelihood information criterion approximation (QIU) values were obtained to select an appropriate working correlation matrix [23].

We included a trend analysis. The model was adjusted for confounders such as BMI categories (underweight, normal, overweight, and obese), age, sex, education level, marital status, income, lifestyle factors, and comorbidities. Odds ratios (ORs) and 95% confidence intervals (CIs) were calculated to determine the strength and significance of associations.

Statistical analyses were performed using STATA/SE version 17 (Stata Corp., College Station, Texas, USA), with the significance level set at 5%. A two-sided test was used, and p-values less than 0.05 were considered statistically significant.

## Results

### General characteristics

The general characteristics of the participants without DM at baseline were examined, and these were observed at two-year intervals (Table 2). In total, 22,016 observations were analyzed.

Among the 22,016 records, 1,351 (6.10%) patients with new-onset DM were observed during the follow-up period. The sample comprised 45% male and 55% female participants. The mean age of all participants was 73.8 years, with a standard deviation of 6.4. The distribution by age group was as follows: 58.90% were young-old adults (65–74 years), 34.40% were old-old adults (75–84 years), and 6.70% were very old adults (≥85 years). The BMI distribution was as follows: 5% were underweight, 45.3% were normal weight, 28.2% were overweight, and 21.5% were obese. The mean absolute HGS was 22.6, with a standard deviation of 8.0, whereas the mean relative HGS was 0.99, with a standard deviation of 0.36. The residential distribution showed 69.5% of the participants resided in rural areas. Marital status indicated that 70.1% of the participants were married. Furthermore, 88.2% of the individuals were nonsmokers and 54.9% were nondrinkers. Regular physical activity, defined as engaging in exercise at least once a week, was observed in 36% of the participants. The mean score for ADL and IADL was 0.07 and 0.43, respectively. The prevalence of diagnosed conditions was as follows: 45.4% of the participants had hypertension, 9.4% had heart disease, 5.5% had cerebrovascular disease, 5.5% had cancer, 3.9% had chronic respiratory disease, and 2.1% had chronic liver disease.

### General characteristics of relative HGS at baseline

Using baseline data from 2006, the general characteristics of relative HGS were examined for each factor (Table 3). Trend tests, including Cochran–Armitage trend tests for sex and major chronic diseases (binary categories) and Jonkheere–Terpstra trend tests for age groups with three or more categories, were conducted to assess the tendency of relative HGS based on each factor. Significant differences in relative HGS were observed between males and females (p for trend < 0.001). Age group analysis showed a decreasing trend in median relative HGS with increasing age (p for trend < 0.001). There was also a trend toward a decrease in median relative HGS as BMI increased (p for trend < 0.001). Individuals with hypertension had a mean relative HGS of 0.92, with a standard deviation of 0.33, whereas those without hypertension had a mean relative HGS of 1.02, with a standard deviation of 0.36; this difference was statistically significant (p < 0.001). Relative HGS exhibited a significant decrease in individuals with cerebrovascular disease (p = 0.02) and a significant increase in those with chronic respiratory disease (p = 0.02).

**Table 2. General characteristics of research participants.**

| Characteristics | 2006 | 2008 | 2010 | 2012 | 2014 | 2016 | 2018 | 2020 | Total |
|---|---|---|---|---|---|---|---|---|---|
| | N = 2,872 | N = 2,754 | N = 2,648 | N = 2,700 | N = 2,699 | N = 2,749 | N = 2,745 | N = 2,849 | N = 22,016 |
| | N(%) | N(%) | N(%) | N(%) | N(%) | N(%) | N(%) | N(%) | N(%) |
| **Gender** | | | | | | | | | |
| Male | 1,309(45.6) | 1,235 (44.8) | 1,206 (45.5) | 1,220 (45.2) | 1,226 (45.4) | 1,221 (44.4) | 1,217 (44.3) | 1,268 (44.5) | 9,902 (45) |
| Female | 1,563(54.4) | 1,519(55.2) | 1,442(54.5) | 1,480(54.8) | 1,473(54.6) | 1,528(55.6) | 1,528(55.7) | 1,581(55.5) | 12,114(55) |
| **Age** | 72.3±5.9 | 72.8±6.0 | 73.3±6.0 | 73.7±6.2 | 73.9±6.3 | 74.3±6.4 | 74.9±6.8 | 75.1±6.9 | 73.8±6.4 |
| 65~74 | 1,967(68.5) | 1,836(66.7) | 1,689(63.8) | 1,614(59.8) | 1,544(57.2) | 1,473(53.6) | 1,380(50.3) | 1,463(51.4) | 12,966(58.9) |
| 75~84 | 796(27.7) | 776(28.2) | 818(30.9) | 926(34.3) | 982(36.4) | 1,059(38.5) | 1,121(40.8) | 1,097(38.5) | 7,575(34.4) |
| ≥ 85 | 109(3.8) | 142(5.2) | 141(5.3) | 160(5.9) | 173(6.4) | 217(7.9) | 244(8.9) | 289(10.1) | 1,475(6.7) |
| **BMI** | 22.7±2.9 | 22.8±2.9 | 22.9±2.9 | 22.9±2.8 | 23.0±2.8 | 23.2±2.8 | 23.3±2.7 | 23.4±2.6 | 23.0±2.8 |
| Thin | 193(6.7) | 151(5.5) | 166(6.3) | 155(5.7) | 136(5) | 120(4.4) | 104(3.8) | 84(2.9) | 1,109(5) |
| Moderate | 1,380(48.1) | 1,332(48.4) | 1,210(45.7) | 1,242(46) | 1,222(45.3) | 1,221(44.4) | 1,162(42.3) | 1,196(42) | 9,965(45.3) |
| Overweight | 761(26.5) | 752(27.3) | 721(27.2) | 755(28) | 758(28.1) | 759(27.6) | 819(29.8) | 887(31.1) | 6,212(28.2) |
| Obese | 538(18.7) | 519(18.8) | 551(20.8) | 548(20.3) | 583(21.6) | 649(23.6) | 660(24) | 682(23.9) | 4,730(21.5) |
| **Absolute HGS** | 22.1±7.8 | 21.8±7.4 | 21.3±7.6 | 22.5±8.6 | 23.3±8.2 | 23.6±8.3 | 23.3±8.0 | 23.2±8.0 | 22.6±8.0 |
| Low | 992(34.5) | 1,006(36.5) | 1,052(39.7) | 966(35.8) | 856(31.7) | 837(30.4) | 806(29.4) | 851(29.9) | 7,366(33.5) |
| Middle | 935(32.6) | 916(33.3) | 840(31.7) | 879(32.6) | 862(31.9) | 859(31.2) | 1,005(36.6) | 1,016(35.7) | 7,312(33.2) |
| High | 992(34.5) | 1,006(36.5) | 1,052(39.7) | 966(35.8) | 856(31.7) | 837(30.4) | 806(29.4) | 851(29.9) | 7,366(33.5) |
| **Relative HGS** | 0.98±0.36 | 0.97±0.34 | 0.94±0.35 | 0.99±0.39 | 1.02±0.37 | 1.03±0.37 | 1.01±0.36 | 1.00±0.36 | 0.99±0.36 |
| Low | 979(34.1) | 980(35.6) | 1,018(38.4) | 930(34.4) | 849(31.5) | 840(30.6) | 844(30.7) | 900(31.6) | 7,340(33.3) |
| Middle | 916(31.9) | 908(33) | 852(32.2) | 905(33.5) | 884(32.8) | 894(32.5) | 982(35.8) | 998(35) | 7,339(33.3) |
| High | 979(34.1) | 980(35.6) | 1,018(38.4) | 930(34.4) | 849(31.5) | 840(30.6) | 844(30.7) | 900(31.6) | 7,340(33.3) |
| **Education level** | | | | | | | | | |
| ≤ Elementary school | 1,987(69.2) | 1,894(68.8) | 1,740(65.7) | 1,654(61.3) | 1,553(57.5) | 1,492(54.3) | 1,371(49.9) | 1,283(45) | 12,974(58.9) |
| Middle school | 315(11) | 317(11.5) | 340(12.8) | 381(14.1) | 426(15.8) | 455(16.6) | 502(18.3) | 534(18.7) | 3,270(14.9) |
| High school | 397(13.8) | 375(13.6) | 393(14.8) | 466(17.3) | 506(18.7) | 564(20.5) | 627(22.8) | 749(26.3) | 4,077(18.5) |
| ≥ College | 173(6) | 168(6.1) | 175(6.6) | 199(7.4) | 214(7.9) | 238(8.7) | 245(8.9) | 283(9.9) | 1,695(7.7) |
| **Residential district** | | | | | | | | | |
| Urban | 2,024(70.5) | 1,870(67.9) | 1,815(68.5) | 1,826(67.6) | 1,844(68.3) | 1,896(69) | 1,926(70.2) | 2,104(73.9) | 15,305(69.5) |
| Rural | 848(29.5) | 884(32.1) | 833(31.5) | 874(32.4) | 855(31.7) | 853(31) | 819(29.8) | 745(26.1) | 6,711(30.5) |
| **Marital status** | | | | | | | | | |
| Married | 1,911(66.5) | 1,879(68.2) | 1,835(69.3) | 1,887(69.9) | 1,919(71.1) | 1,963(71.4) | 1,980(72.1) | 2,050(72) | 15,424(70.1) |
| Not married | 961(33.5) | 875(31.8) | 813(30.7) | 813(30.1) | 780(28.9) | 786(28.6) | 765(27.9) | 799(28) | 6,592(29.9) |
| **Employment** | | | | | | | | | |
| Employed | 560(19.5) | 672(24.4) | 746(28.2) | 716(26.5) | 748(27.7) | 725(26.4) | 758(27.6) | 767(26.9) | 5,692(25.9) |
| Not Employed | 2,312(80.5) | 2,082(75.6) | 1,902(71.8) | 1,984(73.5) | 1,951(72.3) | 2,024(73.6) | 1,987(72.4) | 2,082(73.1) | 16,324(74.1) |
| **Annual household income** | | | | | | | | | |
| Q1 | 1,562(54.4) | 842(30.6) | 678(25.6) | 526(19.5) | 615(22.8) | 451(16.4) | 334(12.2) | 206(7.2) | 5,214(23.7) |
| Q2 | 532(18.5) | 730(26.5) | 766(28.9) | 828(30.7) | 689(25.5) | 769(28) | 803(29.3) | 807(28.3) | 5,924(26.9) |
| Q3 | 416(14.5) | 653(23.7) | 667(25.2) | 732(27.1) | 709(26.3) | 760(27.6) | 737(26.8) | 825(29) | 5,499(25) |
| Q4 | 362(12.6) | 529(19.2) | 537(20.3) | 614(22.7) | 686(25.4) | 769(28) | 871(31.7) | 1,011(35.5) | 5,379(24.4) |
| **Smoking status** | | | | | | | | | |
| Smoker+Former smoker | 471(16.4) | 413(15) | 381(14.4) | 362(13.4) | 294(10.9) | 253(9.2) | 222(8.1) | 196(6.9) | 2,592(11.8) |
| Nonsmoker | 2,401(83.6) | 2,341(85) | 2,267(85.6) | 2,338(86.6) | 2,405(89.1) | 2,496(90.8) | 2,523(91.9) | 2,653(93.1) | 19,424(88.2) |
| **Alcohol consumption** | | | | | | | | | |
| Drinker+ Former drinker | 1,137(39.6) | 1,124(40.8) | 1,155(43.6) | 1,203(44.6) | 1,255(46.5) | 1,314(47.8) | 1,324(48.2) | 1,415(49.7)) | 9,927(45.1) |
| Nondrinker | 1,735(60.4) | 1,630(59.2) | 1,493(56.4) | 1,497(55.4) | 1,444(53.5) | 1,435(52.2) | 1,421(51.8) | 1,434(50.3) | 12,089(54.9) |

(*Continued*)

**Table 2.** (Continued)

| Characteristics | 2006 | 2008 | 2010 | 2012 | 2014 | 2016 | 2018 | 2020 | Total |
|---|---|---|---|---|---|---|---|---|---|
| | N = 2,872 | N = 2,754 | N = 2,648 | N = 2,700 | N = 2,699 | N = 2,749 | N = 2,745 | N = 2,849 | N = 22,016 |
| | N(%) | N(%) | N(%) | N(%) | N(%) | N(%) | N(%) | N(%) | N(%) |
| **Regular Exercise** | | | | | | | | | |
| Yes | 951(33.1) | 878(31.9) | 863(32.6) | 991(36.7) | 946(35.1) | 1,023(37.2) | 984(35.8) | 1,280(44.9) | 7,916(36) |
| No | 1,921(66.9) | 1,876(68.1) | 1,785(67.4) | 1,709(63.3) | 1,753(64.9) | 1,726(62.8) | 1,761(64.2) | 1,569(55.1) | 14,100(64) |
| **ADL index** | 0.14±0.75 | 0.09±0.59 | 0.08±0.61 | 0.07±0.56 | 0.04±0.44 | 0.07±0.52 | 0.06±0.53 | 0.05±0.45 | 0.07±0.56 |
| **IADL index** | 0.64±1.78 | 0.5±1.57 | 0.46±1.57 | 0.35±1.33 | 0.36±1.36 | 0.37±1.42 | 0.37±1.43 | 0.37±1.43 | 0.43±1.5 |
| **Diabetes Mellitus** | | | | | | | | | |
| Yes | 0(0) | 58(2.1) | 115(4.3) | 155(5.7) | 215(8) | 251(9.1) | 268(9.8) | 289(10.1) | 1,351(6.1) |
| No | 2,872(100) | 2,696(97.9) | 2,533(95.7) | 2,545(94.3) | 2,484(92) | 2,498(90.9) | 2,477(90.2) | 2,560(89.9) | 20,665(93.9) |
| **Hypertension** | | | | | | | | | |
| Yes | 1,005(35) | 1,092(39.7) | 1,161(43.8) | 1,241(46) | 1,310(48.5) | 1,364(49.6) | 1,387(50.5) | 1,434(50.3) | 9,994(45.4) |
| No | 1,867(65) | 1,662(60.3) | 1,487(56.2) | 1,459(54) | 1,389(51.5) | 1,385(50.4) | 1,358(49.5) | 1,415(49.7) | 12,022(54.6) |
| **Heart disease** | | | | | | | | | |
| Yes | 193(6.7) | 219(8) | 235(8.9) | 257(9.5) | 276(10.2) | 291(10.6) | 289(10.5) | 299(10.5) | 2,059(9.4) |
| No | 2,679(93.3) | 2,535(92) | 2,413(91.1) | 2,443(90.5) | 2,423(89.8) | 2,458(89.4) | 2,456(89.5) | 2,550(89.5) | 19,957(90.6) |
| **Cerebrovascular disease** | | | | | | | | | |
| Yes | 106(3.7) | 121(4.4) | 135(5.1) | 140(5.2) | 165(6.1) | 183(6.7) | 185(6.7) | 170(6) | 1,205(5.5) |
| No | 2,766(96.3) | 2,633(95.6) | 2,513(94.9) | 2,560(94.8) | 2,534(93.9) | 2,566(93.3) | 2,560(93.3) | 2,679(94) | 20,811(94.5) |
| **Cancer** | | | | | | | | | |
| Yes | 65(2.3) | 91(3.3) | 125(4.7) | 153(5.7) | 177(6.6) | 175(6.4) | 194(7.1) | 239(8.4) | 1,219(5.5) |
| No | 2,807(97.7) | 2,663(96.7) | 2,523(95.3) | 2,547(94.3) | 2,522(93.4) | 2,574(93.6) | 2,551(92.9) | 2,610(91.6) | 20,797(94.5) |
| **Chronic Lung disease** | | | | | | | | | |
| Yes | 107(3.7) | 115(4.2) | 114(4.3) | 105(3.9) | 108(4) | 112(4.1) | 104(3.8) | 99(3.5) | 864(3.9) |
| No | 2,765(96.3) | 2,639(95.8) | 2,534(95.7) | 2,595(96.1) | 2,591(96) | 2,637(95.9) | 2,641(96.2) | 2,750(96.5) | 21,152(96.1) |
| **Chronic Liver disease** | | | | | | | | | |
| Yes | 40(1.4) | 45(1.6) | 52(2) | 56(2.1) | 61(2.3) | 66(2.4) | 66(2.4) | 76(2.7) | 462(2.1) |
| No | 2,832(98.6) | 2,709(98.4) | 2,596(98) | 2,644(97.9) | 2,638(97.7) | 2,683(97.6) | 2,679(97.6) | 2,773(97.3) | 21,554(97.9) |

Data are expressed as means ± standard deviation or number(percentages)

BMI = Body Mass Index, Absolute HGS = Absolute handgrip strength, Relative HGS = Relative handgrip strength,

BMI: Thin = 0–18.4, Moderate = 18.5–22.9, Overweight = 23–24.9, Obese > 25

Absoulte HGS Tertile: Low = 0–18.25, Middle = 18.25–25.75, High = 25.75–84.3

Relative HGS Tertile: Low = 0–0.79, Middle = 0.79–1.13, High = 1.13–4.30

Males

Absolute HGS Tertile: Low = 0–25.80, Middle = 25.80–31.70, High = 31.70–84.30

Relative HGS Tertile: Low = 0–1.13, Middle = 1.13–1.39, High = 1.39–4.30

Females

Absoulte HGS Tertile: Low = 0–15.65, Middle = 15.65–19.95, High = 19.95–49.13

Relative HGS Tertile: Low = 0–0.68, Middle = 0.68–0.86, High = 0.86–2.39

## Selection of the working correlation matrix for the GEE

To select the most suitable model, we compared the QIC and QICu values for the different working correlation matrices (Table 4). The independent working correlation matrix was chosen as the most appropriate for the data because it exhibited the lowest QIC and QICu values. An unstructured and stationary correlation structure is infeasible due to the complexity of its several variables.

**Table 3. General characteristics of relative handgrip strength at baseline.**

| Characteristics | Relative handgrip strength | | | | |
|---|---|---|---|---|---|
| | mean±SD | 1st Tertile | 2nd Tertile | 3rd Tertile | P for trend* |
| | | Low | Middle | High | |
| **Gender** | | | | | |
| Male | 1.26±0.29 | 0.65±0.11 | 0.99±0.09 | 1.4±0.20 | <0.001 |
| Female | 0.75±0.21 | 0.61±0.14 | 0.92±0.09 | 1.21±0.08 | |
| **Age** | | | | | |
| 65~74 | 1.03±0.35 | 0.63±0.13 | 0.95±0.10 | 1.40±0.20 | <0.001 |
| 75~84 | 0.9±0.33 | 0.60±0.14 | 0.94±0.09 | 1.35±0.17 | |
| ≥ 85 | 0.74±0.32 | 0.55±0.17 | 0.97±0.09 | 1.36±0.15 | |
| **Body Mass Index** | | | | | |
| Underweight (< = 18.4) | 1.05±0.43 | 0.59±0.17 | 0.96±0.10 | 1.49±0.29 | <0.001 |
| Normal (18.5–22.9) | 1.04±0.36 | 0.61±0.14 | 0.95±0.09 | 1.42±0.20 | |
| Overweight (23–24.9) | 0.98±0.32 | 0.62±0.13 | 0.93±0.10 | 1.34±0.16 | |
| Obese (>25) | 0.84±0.31 | 0.61±0.13 | 0.94±0.10 | 1.31±0.13 | |
| **Diabetes Mellitus (No)** | | | | | |
| Yes: excluded | | | | | |
| No | 0.98±0.36 | 0.61±0.14 | 0.94±0.10 | 1.39±0.20 | |
| **Hypertension** | | | | | |
| Yes | 0.92±0.33 | 0.61±0.13 | 0.94±0.10 | 1.36±0.17 | <0.001 |
| No | 1.02±0.36 | 0.62±0.14 | 0.95±0.09 | 1.40±0.20 | |
| **Heart disease** | | | | | |
| Yes | 0.93±0.34 | 0.60±0.15 | 0.94±0.09 | 1.34±0.18 | 0.29 |
| No | 0.99±0.36 | 0.61±0.13 | 0.94±0.10 | 1.40±0.20 | |
| **Cerebrovascular disease** | | | | | |
| Yes | 0.95±0.34 | 0.61±0.17 | 0.93±0.10 | 1.40±0.19 | 0.02 |
| No | 0.98±0.36 | 0.61±0.13 | 0.94±0.09 | 1.39±0.20 | |
| **Cancer** | | | | | |
| Yes | 1.09±0.37 | 0.61±0.20 | 0.99±0.09 | 1.40±0.19 | 0.64 |
| No | 0.98±0.36 | 0.61±0.13 | 0.94±0.10 | 1.39±0.20 | |
| **Chronic Lung disease** | | | | | |
| Yes | 1.01±0.36 | 0.61±0.11 | 0.94±0.09 | 1.35±0.19 | 0.02 |
| No | 0.98±0.36 | 0.61±0.14 | 0.94±0.10 | 1.39±0.20 | |
| **Chronic Liver disease** | | | | | |
| Yes | 1.06±0.4 | 0.61±0.14 | 0.94±0.10 | 1.39±0.19 | 0.35 |
| No | 0.98±0.36 | 0.59±0.14 | 0.92±0.10 | 1.41±0.25 | |
| **Total** | 0.98±0.36 | 0.61±0.14 | 0.94±0.10 | 1.39±0.20 | |

Data are expressed as means ± standard deviation or relative handgrip strength tertiles.

* Since gender and chronic diseases are two categorical variables, a trend test was performed using the Cochran-Armitage trend test to calculate p for trend value. For age groups, a trend test was performed using the Jonkheere-Terpstra trend test, which corresponds to cases where there are three or more categories, and p for trend value was calculated.

## Analysis of the effect of relative HGS on new-onset DM

After selecting the working correlation matrix, we performed GEE analysis using the logit function to investigate the impact of relative HGS and other control factors on new-onset DM (Table 5). In this process, BMI was not a control variable, but was analyzed by including it in

**Table 4. Generalized estimating equation model covariance structure analysis.**

|  | Independent | Autoregressive 1 | Exchangeable | Unstructured | Stationary 1 |
|---|---|---|---|---|---|
| QIC | 9229.528 | 9401.661 | 9326.328 | Impossible to estimate | Impossible to estimate |
| QICu | 9150.794 | 9312.264 | 9267.156 | Impossible to estimate | Impossible to estimate |

QIC = Quasi-likelihood information criterion; QICu = Quasi-likelihood information criterion approximation.

the GEE model to evaluate collinearity. In male elderly, compared with the lowest relative HGS group, the odds of developing DM were reduced by 0.87-fold (odds ratio [OR] 0.87, 95% CI 0.61–1.23, p = 0.419) in the intermediate relative HGS group and by 0.82-fold in the high relative HGS group. (OR 0.82, 95% CI 0.56–1.18, p = 0.281). Among femlae elderly individuals, compared with the lowest relative HGS group, the odds of developing DM were reduced by 0.82-fold (odds ratio [OR] 0.82, 95% CI 0.66–1.03, p = 0.087) in the intermediate relative HGS group and by 0.79-fold in the high relative HGS group. (OR 0.79, 95% CI 0.50–1.25, p = 0.306). However, these differences were not statistically significant.

Furthermore, in the GEE model analysis, in addition to relative HGS, age, BMI, overweight or obese, household income, alcohol consumption, hypertension (OR = 3.59), and chronic liver disease (OR = 2.17) significantly contributed to DM incidence in the overall elderly population. In the elderly male population, the probability of DM incidence increased significantly with age (Old-old(75~84) OR = 3.58, Oldest-old($\geq$ 85) OR = 2.73), BMI overweight (OR = 1.42) or obese (OR = 1.53), diagnosis of hypertension (OR = 3.96), and diagnosis of chronic liver disease(OR = 2.76). In the elderly female population, the probability of DM incidence increased significantly with age (OR = 2.65 for Old-old(75~84), OR = 2.33 for Oldest-old($\geq$ 85)), BMI obese (OR = 1.49), being in the second quartile of annual household income (OR = 1.47), and diagnosis of hypertension (OR = 3.35). These results also indicate that BMI as a more robust predictor of DM onset compared to relative HGS.

## Trend test for DM incidence according to relative HGS

Logistic regression analysis was performed to obtain the ORs, and a trend test was performed to examine the trend in DM incidence according to relative HGS. The OR of DM onset according to relative HGS was calculated using logistic regression analysis adjusted for BMI and all control variables, and the OR of DM onset were calculated using the Mantel–Haenszel test. A stratified analysis by sex was performed (Table 6).

In the total population, higher relative HGS was associated with a lower risk of DM, with odds ratios (OR) for the 2nd and 3rd tertiles being 0.71 (95% CI: 0.62–0.81) and 0.65 (95% CI: 0.57–0.74) respectively in the unadjusted model (P for trend < 0.0001). This association remained significant in the adjusted model, with ORs of 0.83 (95% CI: 0.66–1.03) and 0.72 (95% CI: 0.52–1.00) (P for trend = 0.0048). Stratified analysis by gender showed that in males, the unadjusted ORs for the 2nd and 3rd tertiles were 0.74 (95% CI: 0.55–0.98) and 0.48 (95% CI: 0.37–0.63) respectively (P for trend < 0.0001), and the adjusted ORs were 0.68 (95% CI: 0.29–1.56) and 0.60 (95% CI: 0.31–1.14) (P for trend = 0.0189). Among females, the unadjusted ORs were 0.56 (95% CI: 0.48–0.67) and 0.48 (95% CI: 0.32–0.71) (P for trend < 0.0001), but the adjusted model did not show a significant trend, with ORs of 0.85 (95% CI: 0.66–1.09) and 0.68 (95% CI: 0.34–1.34) (P for trend = 0.1087). These findings suggest that higher relative HGS is generally associated with a lower likelihood of DM diagnosis, particularly in males, even after adjusting for multiple confounders.

**Table 5. Generalized estimation equation model analysis of diabetes mellitus incidence according to relative handgrip strength.**

| | DM | | | | | |
| --- | --- | --- | --- | --- | --- | --- |
| | Total | | Male | | Female | |
| | OR (95%CI) | *P*-value | OR (95%CI) | *P*-value | OR (95%CI) | *P*-value |
| **Relative Handgrip Strength** | | | | | | |
| Low | 1 | | 1 | | 1 | |
| Middle | 0.88 (0.73,1.07) | 0.197 | 0.87 (0.61,1.23) | 0.419 | 0.82 (0.66,1.03) | 0.087 |
| High | 0.89 (0.69,1.13) | 0.329 | 0.82 (0.56,1.18) | 0.281 | 0.79 (0.5,1.25) | 0.306 |
| **Age** | | | | | | |
| Young-old (65~74) | 1 | | 1 | | 1 | |
| Old-old (75~84) | 3.12 (2.63,3.7) | <.0001 | 3.58 (2.81,4.57) | <.0001 | 2.65 (2.09,3.35) | <.0001 |
| Oldest-old ($\geq$ 85) | 2.65 (1.91,3.66) | <.0001 | 2.73 (1.66,4.49) | <.0001 | 2.33 (1.51,3.59) | <.0001 |
| **Body Mass Index** | | | | | | |
| Underweight (< = 18.4) | 0.73 (0.46,1.17) | 0.198 | 0.79 (0.41,1.52) | 0.478 | 0.73 (0.38,1.42) | 0.358 |
| Normal (18.5–22.9) | 1 | | 1 | | 1 | |
| Overweight (23–24.9) | 1.32 (1.05,1.65) | 0.015 | 1.42 (1.02,1.99) | 0.039 | 1.21 (0.9,1.63) | 0.209 |
| Obese (>25) | 1.54 (1.2,1.99) | 0.001 | 1.53 (1.03,2.27) | 0.034 | 1.49 (1.08,2.07) | 0.016 |
| **Education level** | | | | | | |
| $\leq$Elementary school | 1 | | 1 | | 1 | |
| Middle school | 0.87 (0.63,1.22) | 0.419 | 0.95 (0.61,1.48) | 0.817 | 0.75 (0.45,1.26) | 0.283 |
| High school | 1.14 (0.85,1.53) | 0.375 | 1.24 (0.83,1.85) | 0.285 | 0.9 (0.56,1.44) | 0.659 |
| $\geq$College | 1.02 (0.65,1.6) | 0.932 | 1 (0.59,1.7) | 0.993 | 1.19 (0.43,3.28) | 0.732 |
| **Annual household income** | | | | | | |
| Q1 | 1 | | 1 | | 1 | |
| Q2 | 1.4 (1.13,1.74) | 0.002 | 1.35 (0.98,1.86) | 0.071 | 1.47 (1.1,1.96) | 0.01 |
| Q3 | 1.32 (1.02,1.7) | 0.032 | 1.41 (0.98,2.05) | 0.065 | 1.26 (0.89,1.8) | 0.193 |
| Q4 | 1.07 (0.8,1.43) | 0.63 | 0.89 (0.58,1.38) | 0.602 | 1.31 (0.9,1.9) | 0.164 |
| **Residential district** | | | | | | |
| Urban | 1 | | 1 | | 1 | |
| Rural | 0.89 (0.69,1.15) | 0.362 | 0.69 (0.47,1.01) | 0.058 | 1.07 (0.77,1.49) | 0.697 |
| **Marital status** | | | | | | |
| Married | 1 | | 1 | | 1 | |
| Not married | 1.21 (0.96,1.54) | 0.109 | 1.34 (0.87,2.06) | 0.189 | 1.26 (0.93,1.71) | 0.141 |
| **Smoking status** | | | | | | |
| Nonsmoker | 1 | | 1 | | 1 | |
| Smoker + Former smoker | 1.1 (0.79,1.53) | 0.583 | 1.13 (0.79,1.62) | 0.504 | 0.8 (0.3,2.15) | 0.656 |
| **Alcohol consumption** | | | | | | |
| Nondrinker | 1 | | 1 | | 1 | |
| Drinker + Former drinker | 1.36 (1.08,1.71) | 0.009 | 1.34 (0.89,2.02) | 0.161 | 1.2 (0.84,1.71) | 0.309 |
| **Regular Exercise** | | | | | | |
| No | 1 | | 1 | | 1 | |
| Yes | 1.16 (0.99,1.38) | 0.074 | 1.22 (0.95,1.55) | 0.117 | 1.11 (0.87,1.4) | 0.395 |
| **ADL index (0~7)** | 0.98 (0.87,1.1) | 0.733 | 0.95 (0.8,1.14) | 0.614 | 0.99 (0.84,1.17) | 0.932 |
| **IADL index (0~10)** | 1.02 (0.97,1.08) | 0.398 | 1.06 (0.97,1.15) | 0.195 | 1 (0.93,1.08) | 0.991 |
| **Hypertension** | | | | | | |
| No | 1 | | 1 | | 1 | |
| Yes | 3.59 (2.83,4.57) | <0.0001 | 3.96 (2.79,5.62) | <.0001 | 3.35 (2.41,4.65) | <.0001 |
| **Heart disease** | | | | | | |
| No | 1 | | 1 | | 1 | |

(*Continued*)

**Table 5.** (Continued)

| | DM | | | | | |
| --- | --- | --- | --- | --- | --- | --- |
| | Total | | Male | | Female | |
| | OR (95%CI) | P-value | OR (95%CI) | P-value | OR (95%CI) | P-value |
| Yes | 1.07 (0.79,1.43) | 0.668 | 0.85 (0.54,1.34) | 0.486 | 1.28 (0.87,1.88) | 0.218 |
| **Cerebrovascular disease** | | | | | | |
| No | 1 | | 1 | | 1 | |
| Yes | 1.26 (0.88,1.79) | 0.202 | 1.06 (0.65,1.73) | 0.801 | 1.49 (0.88,2.51) | 0.134 |
| **Cancer** | | | | | | |
| No | 1 | | 1 | | 1 | |
| Yes | 1.13 (0.77,1.65) | 0.523 | 1.05 (0.63,1.74) | 0.86 | 1.28 (0.72,2.28) | 0.395 |
| **Chronic Lung disease** | | | | | | |
| No | 1 | | 1 | | 1 | |
| Yes | 1.15 (0.73,1.8) | 0.548 | 0.94 (0.49,1.81) | 0.849 | 1.47 (0.8,2.72) | 0.217 |
| **Chronic Liver disease** | | | | | | |
| No | 1 | | 1 | | 1 | |
| Yes | 2.17 (1.23,3.83) | 0.008 | 2.76 (1.29,5.88) | 0.009 | 1.44 (0.56,3.67) | 0.45 |
| **QIC** | 9200 | | 9257.8 | | 9301.9 | |

Stratified analysis by gender was performed. Values are presented as odds ratios (95% confidence interval).

GEE was used to investigate the association between degree of grip strength and DM.

Control variables: BMI*, age, education level, residential district, marital status, annual household income, smoking status, alcohol consumption status, regular exercise, ADL index, IADL index, diagnosis of hypertension, diagnosis of heart disease, diagnosis of cerebrovascular disease, diagnosis of cancer, diagnosis of chronic lung disease diagnosis, and diagnosis of chronic liver disease.

*Although BMI is not a control variable, analysis was performed by including it in the GEE model to evaluate collinearity.

**Table 6. Trends analysis in diabetes mellitus diagnosis according to relative handgrip strength.**

| Subjects | Model | Relative handgrip strength tertile group | | | P for trend |
| --- | --- | --- | --- | --- | --- |
| | | 1st tertile | 2nd tertile | 3rd tertile | |
| | | OR (95% CI) | OR (95% CI) | OR (95% CI) | |
| **Total** | Unadjusted | 1 | 0.71 (0.62–0.81) | 0.65 (0.57–0.74) | <0.0001 |
| | Adjusted* | 1 | 0.83 (0.66–1.03) | 0.72 (0.52–1.00) | 0.0048 |
| **Male** | Unadjusted | 1 | 0.74 (0.55–0.98) | 0.48 (0.37–0.63) | <0.0001 |
| | Adjusted* | 1 | 0.68 (0.29–1.56) | 0.60 (0.31–1.14) | 0.0189 |
| **Female** | Unadjusted | 1 | 0.56 (0.48–0.67) | 0.48 (0.32–0.71) | <0.0001 |
| | Adjusted* | 1 | 0.85 (0.66–1.09) | 0.68 (0.34–1.34) | 0.1087 |

Stratified analysis by gender was performed.

Values are presented as odds ratios (95% confidence interval).

*Adjusted Model: adjusted for age, body mass index (Underweight/Normal (reference) /Overweight / Obese), education level, residential district, marital status, annual household income, smoking status, alcohol consumption status, regular exercise, ADL index, IADL index, diagnosis of hypertension, diagnosis of heart disease, diagnosis of cerebrovascular disease, diagnosis of cancer, diagnosis of chronic lung disease diagnosis, and diagnosis of chronic liver disease.

The odds ratio of diabetes diagnosis according to relative handgrip strength was obtained through logistic regression analysis.

Afterwards, the odds ratios of diabetes mellitus diagnosis were tested using the Mantel–Haenszel test.

## Discussion

This study investigated the longitudinal association between relative HGS and DM incidence based on a nationally representative panel survey of elderly Koreans. Using data from the KLoSA conducted from 2006 to 2020, the ORs for new-onset DM in Korean individuals aged ≥65 years were examined in relation to relative HGS. The results showed a decrease in the odds of DM incidence in groups with higher relative HGS than those in the group with the lowest HGS among older adults. However, the results did not achieve statistical significance, while overweight and obesity groups increased in the odds of DM incidence. Our findings suggests that BMI accounts for new DM incidence in the majority. However, even after adjusting for BMI, the trend test showed that high relative HGS tended to reduce the odds of newly onset DM, especially in men. In addition to managing obesity and chronic conditions, such as hypertension and liver disease, educational interventions promoting resistance exercises to enhance HGS might be required for DM prevention.

In a prospective cohort study of 66,100 Europeans aged ≥50 years without DM using data from the Survey of Health, Ageing and Retirement in Europe (SHARE), relative HGS was found to be a better predictor of new-onset DM than absolute HGS in older European populations [12]. In addition, although it was a study conducted only on absolute HGS, the HELIUS (Healthy Life in an Urban Setting) study in Amsterdam included 2,086 Dutch people, 2,216 South Asian Surinamese, 2,084 African Surinamese, 1,786 Ghanaians, 2,223 Turks, and Moroccans; in this study, 12,594 participants (2,199 participants) were tested, and a significant difference between HGS and type 2 DM was observed among the races. However, all population groups had an inverse association regardless of race, male sex, female sex, or BMI (OR 0.95; 95% CI 0.92–0.97) [24]. In a previous study conducted on middle-aged and older adults who participated in the National Health and Nutrition Examination Survey in the United States and the Health and Retirement Longitudinal Study in China, the concept of normalized HGS (NGS) was calculated by dividing absolute HGS only by body weight. However, it was confirmed that each 0.05 decrease in NGS was independently associated with a 1.49-fold (95% CI 1.42–1.56) increase in the odds of DM in Americans and a 1.17-fold (95% CI 1.11–1.23) increase in the Chinese [25].

Obesity, which corresponds to a high BMI, is an important predictor of developing DM. However, according to a recent study, it has been reported that HGS, which represents upper body strength, is more closely related to mortality or metabolic diseases including Type 2 DM than lower body strength [26]. In a study on leaner Japanese Americans, it was found that in people with a BMI in the bottom 25%, the greater the HGS, the lower the risk of developing DM [27]. There are cases where the BMI is high but the muscle strength is high due to hard exercise, and it is difficult for absolute HGS to consider these cases. Therefore, in this study, relative HGS was defined as an independent variable to determine whether the risk of DM is high when low muscle strength and high BMI coexist.

In the current study, the sex-stratified analysis revealed differences between male and female elderly individuals. Notably, the impact of chronic liver disease on the relationship between HGS and DM incidence was significant in older males, but not in older females. This sex difference is presumed to be related to the influence of testosterone on muscle strength [28]. Research has suggested that the increase in visceral fat and decrease in muscle strength associated with testosterone deficiency induce the release of various inflammatory cytokines, such as interleukin-1 beta (IL-1β), interleukin-6 (IL-6), and tumor necrosis factor-alpha (TNF-α). These differences may contribute to the sex disparities in DM incidence [29–32].

The main mechanism by which a trend to decrease in relative HGS leads to an increase in DM has not been clearly elucidated yet. However, decrease in relative HGS may decrease

muscle quality and ultimately cause insulin resistance [33]. In other words, it is thought that in individuals with low relative HGS, a decrease in the movement of GLUT 4, a skeletal muscle sugar transport protein, decreases the glucose metabolic ability of the skeletal muscle, develops insulin resistance, and increases DM [34, 35]. However, since the clear mechanism has not been elucidated yet, further investigation through related laboratory studies is warranted.

This study has some limitations. First, there were no internationally standardized criteria for relative HGS. Previous studies on HGS have divided participants' relative HGS into dichotomies, tertiles, quartiles, and so on, leading to varying ORs for DM incidence based on relative HGS. In the present study, categorizing the relative HGS into tertiles may have introduced variability into the results. Therefore, additional research on the globally accepted standards for relative HGS is necessary to predict the future incidence of DM. Second, the height and weight variables were based on reported values, rather than direct physical measurements, potentially resulting in inaccurate BMI values and unreliable research outcomes. Measured anthropometry provides a more reliable tool to assess obesity prevalence. Previous research has shown that individuals tend to over- or under-report weight and height depending on gender and age, although Asian studies have shown lower bias compared to other continents [36, 37]. However, the KLoSA data used in this study captures various aspects of an aging society and builds data that allows for interdisciplinary research in various fields, allowing for international comparison with countries that are already conducting panel surveys on middle-aged and elderly people, such as the United States and Europe. Since this is this study is a real world study that reflects reality and KLoSA data is a large-scale data representative of Korea's elderly population that was intended to produce as much data as possible, the impact of such biases is expected to be relatively small. Third, because this study focused on the elderly population in Korea, it may be challenging to generalize the results to other races or age groups. Therefore, future research should consider a more diverse population in relation to the topic. Fourth, while a typical DM diagnosis requires biochemical data, such as glycated hemoglobin or fasting blood glucose levels, the KLoSA survey did not include blood tests, thereby representing a limitation. Finally, there is a possibility of misclassification bias owing to participants incorrectly reporting their DM diagnoses.

However, this study is significant as the first longitudinal analysis of Korean seniors demonstrating that a higher relative HGS is associated with a decreased risk of DM incidence. Based on a nationally representative cohort of Koreans aged ≥65 years, the study's data strengthens its applicability to the general population. Additionally, this study differs from previous research in that it considers and adjusts for potential variations in the relationship between HGS and DM incidence due to sex- or age-related differences in the physiological characteristics of HGS. Moreover, it directly measures HGS in Asian populations, particularly in Korean seniors. This study used cohort data collected over 14 years to provide robust statistical power to infer a causal relationship between relative HGS and DM incidence.

Based on the results of this study, recommendations for DM prevention in Korean seniors include resistance exercises for muscle strengthening, proper nutritional intake, and weight reduction to lower the BMI. These findings will improve the health and quality of life of the elderly population, aiding in the prevention of DM and its associated complications. Specifically, the study outcomes are expected to enhance understanding and inform medical professionals and policymakers in developing policies and intervention programs for senior health improvement. Investigating the association between relative HGS and DM allows healthcare experts to develop personalized treatment plans for older adults, thereby enhancing their health and quality of life.

In conclusion, this study provides valuable insights into the association between relative HGS and DM incidence in the elderly population. These findings suggest that maintaining or

improving relative HGS through interventions, such as abstinence and resistance exercises and decreasing BMI in overweight and obese, may be an effective strategy for DM prevention, particularly in older adults.

## Supporting information

**S1 Data. Raw dataset collected from the KLoSA from 2006 to 2020.** This dataset includes all the variables recorded during the study period.
(CSV)

**S2 Data. This dataset includes detailed information for 22,016 individuals.** The data encompasses various demographic, clinical, and outcome variables collected during the study period.
(XLSX)

**S3 Data. These files contain the STATA program codes used in the study.** Each file reflects the STATA program codes corresponding to different variations of this research design.
(ZIP)

## Author Contributions

**Conceptualization:** Yeo Ju Sohn, Eun Jee Chang.

**Data curation:** Yeo Ju Sohn.

**Formal analysis:** Yeo Ju Sohn, Eun Jee Chang, Sungchan Kang.

**Investigation:** Yeo Ju Sohn.

**Methodology:** Yeo Ju Sohn, Sungchan Kang.

**Project administration:** Yeo Ju Sohn.

**Software:** Sungchan Kang.

**Supervision:** Hong Soo Lee, Hee Cheol Kang, Sang Wha Lee, Kyung Won Shim.

**Validation:** Hong Soo Lee, Hasuk Bae, Sang Wha Lee, Kyung Won Shim.

**Visualization:** Yeo Ju Sohn.

**Writing – original draft:** Yeo Ju Sohn.

**Writing – review & editing:** Yeo Ju Sohn, Hong Soo Lee, Hasuk Bae, Hee Cheol Kang, Hyejin Chun, Insun Ryou, Eun Jee Chang, Sang Wha Lee, Kyung Won Shim.

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
