## [Decision Letter · Decision Letter 0]

27 Mar 2024

PONE-D-24-01896Association of relative handgrip strength on the development of diabetes mellitus in elderly Koreans.PLOS ONE

Dear Dr. SOHN,

Thank you for submitting your manuscript to PLOS ONE. After careful consideration, we feel that it has merit but does not fully meet PLOS ONE’s publication criteria as it currently stands. Therefore, we invite you to submit a revised version of the manuscript that addresses the points raised during the review process.

We look forward to receiving your revised manuscript.

Kind regards,

Mario Ulises Pérez-Zepeda, M.D., Ph.D.

Academic Editor

PLOS ONE

Journal Requirements:

3. Please include your tables as part of your main manuscript and remove the individual files. Please note that supplementary tables (should remain/ be uploaded) as separate "supporting information" files.

Reviewers' comments:

Reviewer's Responses to Questions

**Comments to the Author**

1. Is the manuscript technically sound, and do the data support the conclusions?

Reviewer #1: Yes

Reviewer #2: Yes

2. Has the statistical analysis been performed appropriately and rigorously? 

Reviewer #1: No

Reviewer #2: Yes

3. Have the authors made all data underlying the findings in their manuscript fully available?

Reviewer #1: Yes

Reviewer #2: Yes

4. Is the manuscript presented in an intelligible fashion and written in standard English?

Reviewer #1: Yes

Reviewer #2: Yes

5. Review Comments to the Author

**Reviewer #1:** This manuscript explore the role of relative handgrip strength for predicting the incidence of diabetes diagnosis in the elder population. It raised the interest of readers about the significance of handgrip strength on preventing diabetes in the elderly. However, several questions need to be elucidated clearly.

First, please specify the BMI categories in the methodology with adequate reference citation. The ordinary terms of underweight, normal, overweight, and obesity were suggested, rather than thin, moderate, overweight, and obesity. And the criteria for Asian population must have different interpreting cut off value by WHO.

Second, BMI had a main impact on the calculated relative HGS. For example, those obese population would mostly have lower relative HGS after calculation, while obesity is the critical predictor for diabetes incidence. There should be a clear rationale for relative HGS, and it was suggested to have a comparison with absolute HGS, or HGS adjusted for height square. Current calculation of relative HGS may have a main concern of collinearity with BMI. And please have a discussion on the relationship between relative HGS and BMI in depth.

Third, it was suggested to present BMI distribution in both table 3 and 5. There would be a strong interaction between relative HGS and BMI, while BMI might play a more critical role on predicting diabetes incidence. Therefore, it was suggested to include BMI for analysis, and to create a interaction term between BMI and relative HGS in GEE model. Otherwise, there should be a clear rationale why BMI was not included.

Final, some minor errors were suggested to check again. Such as the flow chart for participants inclusion, 63125 minus 29514 leaves 33611, not 33701. And the number should be presented as mean +/- SD in each tertile group in table 3.

**Reviewer #2:** Manuscript Title: Association of relative handgrip strength on the development of diabetes mellitus in elderly Koreans.

The study investigates the correlation between relative handgrip strength (HGS) and the likelihood of developing diabetes in a Korean population cohort of 22,016 older adults. The research presents valuable insights into the potential relationship between relative HGS and diabetes risk, although several methodological and interpretational aspects require further consideration.

Major comments:

• The authors should compare the predictive abilities of relative HGS, absolute HGS, and BMI in predicting diabetes development to ascertain any additional value provided by relative HGS over other measurements. Furthermore, it would be beneficial to assess whether relative HGS augments the predictive capacity of measured levels of physical activity in predicting new-onset diabetes among older adults.

• The study limitations acknowledge using self-reported height and weight variables, which may introduce recall bias, particularly among older adults. Given the centrality of these variables to the calculation of relative HGS, the authors should explore alternative methods, such as using measured height and weight or employing absolute HGS. Additionally, utilizing self-reporting to diagnose new-onset diabetes may introduce unreliability; therefore, including cases with documented medical records or employing more reliable alternatives is advisable. Incorporating only cases with reliable records, although potentially reducing the sample size, would improve the quality and reliability of the evidence obtained.

• Clarification is needed regarding how medical comorbidities were diagnosed, as this could impact the interpretation of the study findings.

• Details regarding the dynamometer used for measuring HGS should be provided to ensure transparency and replicability of the study methodology.

• The authors should mention whether cases with painful hand diseases or neurological disorders that could affect HGS were excluded from the analysis, as their inclusion could confound the results.

• Considering the complexity of the Generalized Estimating Equation (GEE) model covariance structure analysis, engaging an expert statistical reviewer to validate the approach and interpretation would strengthen the study's methodological rigor.

Minor Comments:

• The introduction appears lengthy and contains redundancy. Streamlining the introduction by eliminating redundant information would enhance its clarity and conciseness.

• The definition of relative HGS should be removed from the introduction and added to the methods section.

Overall, the study presents a valuable contribution to exploring the correlation between relative HGS and diabetes risk. Addressing the aforementioned comments would enhance the methodological robustness of the research findings.

6. PLOS authors have the option to publish the peer review history of their article (what does this mean?). If published, this will include your full peer review and any attached files.

Reviewer #1: **Yes: **LiangYu Chen

Reviewer #2: **Yes: **Abdelrahman Hassan Abdallah Noureldin

---

## [Author Response · Author response to Decision Letter 0]

29 May 2024

# Response to Reviewers

May/24/2024

Dear Editor:

We respectfully resubmit the revised manuscript entitled: “ Association of relative handgrip strength on the development of diabetes mellitus in elderly Koreans.” The paper was coauthored by Yeo Ju Sohn, Hong Soo Lee, Hasuk Bae, Hee Cheol Kang, Hyejin Chun, Insun Ryou, Eun Jee Chang, Sungchan Kang, Sang Wha Lee, Kyung Won Shim. 

The authors greatly appreciated the careful and thoughtful review from reviewers. We are very pleased to have the opportunity to edit our manuscript. We have read the comments from the editors and reviewers carefully and have revised our paper thoroughly based on these comments. Point-by-point responses to the criticisms and suggestions are described below, and the modified contents are presented in highlight in the revised version (Marked-up Manuscript file).

In this study, we explored the relationship between relative handgrip strength (HGS) and diabetes mellitus(DM) in an elderly Korean population to recognize the importance of simultaneously evaluating the risks of low muscle strength and increased body mass index. We believe that our study significantly contributes to the literature because our findings suggest that maintaining or improving HGS through interventions such as abstinence and resistance exercises may be an effective strategy for DM prevention, particularly in older adults. 

Further, we believe that this paper will be of interest to the readership of your journal because we show that relative HGS can be a useful predictive factor of DM incidence and can be utilized as a readily measurable surrogate for physical function and muscle strength in predicting DM in clinical settings.

This manuscript has not been published or presented elsewhere in part or entirety and is not under consideration by another journal. All study participants provided informed consent, and the appropriate ethics review board approved the study design. We have read and understood your journal’s policies and believe that neither the manuscript nor the study violates these. There are no conflicts of interest to declare.

We have read and understood your journal’s policies, and we believe that neither the manuscript nor the study violates any of these. There are no conflicts of interest to declare.

We kindly ask you to review the amendments, and we eagerly anticipate your favorable consideration in the future. Thank you for your consideration. I look forward to hearing from you.

Sincerely,

Yeo Ju Sohn 

Department of Family Medicine, Ewha Womans University Seoul Hospital, Ewha Womans University College of Medicine, Gonghang-daero 260, Gangseo-gu, Seoul, Korea

Tel: +82-10-2286-2579, Fax: +82-2-2654-2439 E-mail: yeoju822@naver.com

ORCID : https://orcid.org/0000-0002-1196-4011

Reviewer #1: This manuscript explore the role of relative handgrip strength for predicting the incidence of diabetes diagnosis in the elder population. It raised the interest of readers about the significance of handgrip strength on preventing diabetes in the elderly. However, several questions need to be elucidated clearly.

First, please specify the BMI categories in the methodology with adequate reference citation. The ordinary terms of underweight, normal, overweight, and obesity were suggested, rather than thin, moderate, overweight, and obesity. And the criteria for Asian population must have different interpreting cut off value by WHO.

We sincerely appreciate the valuable feedback provided by the reviewer. Based on their insightful comments, we have made significant revisions to our study. We have revised the Introduction as follows.

We defined cutoffs for underweight (<18.5 kg/m2), normal (18.5–22.9 kg/m2), overweight (23.0–24.9 kg/m2), and obesity(≥25.0 kg/m2) based on the WHO Asia-Pacific regional guidelines.[1] 

Second, BMI had a main impact on the calculated relative HGS. For example, those obese population would mostly have lower relative HGS after calculation, while obesity is the critical predictor for diabetes incidence. There should be a clear rationale for relative HGS, and it was suggested to have a comparison with absolute HGS, or HGS adjusted for height square. Current calculation of relative HGS may have a main concern of collinearity with BMI. And please have a discussion on the relationship between relative HGS and BMI in depth. 

We sincerely appreciate the valuable feedback provided by the reviewer. Based on their insightful comments, we have made significant revisions to our study. We have revised the Introduction as follows.

HGS is an indicator with a significant correlation with sarcopenia, which refers to muscle loss with age, and serves as a basic indicator to evaluate overall physical ability and muscle function, especially in the elderly population. According to previous studies, it has been revealed that there is a correlation between BMI and HGS. [2] [3] Additionally, it has been reported that there is a deep correlation between a decrease in muscle mass and a decrease in HGS grip strength. [4] Previous studies have demonstrated the usefulness of absolute handgrip strength in the identification of various health problems and its potential as a new vital sign across the lifespan, but have shown conflicting results. [5] The confounding effect of body size was thought to be one of the causes. In other words, previous studies conducted only with the concept of absolute HGS showed inconsistent results, and this was the result of using absolute HGS as an indicator of muscle strength without body mass correction, as absolute grip strength is closely related to body mass index. Relative grip strength has therefore been recommended as a better metric to take into account the effects of both body mass and muscle strength, and this measure helps account for differences in hand grip strength that may be influenced by an individual's overall size. Various previous studies have demonstrated that relative HGS may be beneficial in predicting cardiovascular biomarkers, metabolic profile, and risk of other cardiometabolic disorders. [6] [7] [8] In addition, there are previous research results showing that relative HGS can predict new-onset DM better than absolute HGS. A study conducted in middle-aged and older adults in Europe also found that low HGS was an independent predictor of new-onset diabetes risk, suggesting that relative HGS had a slightly higher predictive ability for future diabetes than absolute HGS in people aged 50 years or older. have emphasized that screening for low HGS may be of value in preventing diabetes.[7] 

We sincerely appreciate the valuable feedback provided by the reviewer. Based on their insightful comments, we have made significant revisions to our study. We have revised the discussion as follows.

Obesity, which corresponds to a high BMI, is an important predictor of developing diabetes mellitus, but there also exist metabolically obese normal weight phenotype or Obesity paradox. Some people have a high BMI but still have a high muscle mass, and although obesity increases various cardiovascular disease risk factors, patients with various types of CVD sometimes have a better prognosis if they are classified as overweight or obese. This phenomenon may be viewed as being due to the “low-fat paradox,” as individuals classified as normal or underweight have a poor prognosis for cardiovascular disease. [9] There is also visceral fat type obesity with a low BMI but high visceral fat, or obesity with only a high waist circumference, and it is difficult to sufficiently predict diabetes using the BMI indicator alone. In contrast to lower body weight, high muscle strength levels or high HGS in old age are known to play a protective role against the risk of premature death or cardiometabolic disease such as diabetes mellitus. According to a recent study, it has been reported that grip strength, which represents upper body strength, is more closely related to mortality or metabolic diseases including Type 2 Diabetes Mellitus than lower body strength. [10] In a study on leaner Japanese Americans, it was found that in people with a BMI in the bottom 25%, the greater the grip strength, the lower the risk of developing diabetes mellitus. [11] There are cases where the BMI is high but the muscle strength is high due to hard exercise, and it is difficult for absolute HGS to consider these cases. Therefore, in this study, relative grip strength was defined as an independent variable to determine whether the risk of diabetes mellitus is high when low muscle strength and high BMI coexist. 

Third, it was suggested to present BMI distribution in both table 3 and 5. There would be a strong interaction between relative HGS and BMI, while BMI might play a more critical role on predicting diabetes incidence. Therefore, it was suggested to include BMI for analysis, and to create a interaction term between BMI and relative HGS in GEE model. Otherwise, there should be a clear rationale why BMI was not included.

We sincerely appreciate the valuable feedback provided by the reviewer. Based on their insightful comments, we have made significant revisions to our study. We have revised Tables 3 and 5 and Results according to the reviewer’s recommendations. As suggested, we have incorporated BMI into our analysis. Specifically, we created an interaction term between BMI and relative handgrip strength (HGS) in the generalized estimating equation (GEE) model. Concerns about collinearity between BMI and relative HGS were noted, as this relationship led to increased p-values. However, it is noteworthy that despite this collinearity, a consistent trend remains: higher handgrip strength is associated with a reduced odds ratio (OR) for newly-onset diabetes mellitus (DM) in both men and women. Therefore, while acknowledging the collinearity issue, we believe that relative HGS serves as a novel surrogate marker for predicting DM risk and contributes to our understanding of preventive strategies for older adults. Once again, we express our gratitude for the constructive feedback, which has undoubtedly strengthened the scientific rigor of our study. We remain committed to advancing our understanding of DM risk factors and preventive measures.

Final, some minor errors were suggested to check again. Such as the flow chart for participants inclusion, 63125 minus 29514 leaves 33611, not 33701. And the number should be presented as mean +/- SD in each tertile group in table 3. 

We sincerely appreciate the valuable feedback. It seems there was an error in the calculation process. For those under 65 years of age, the correct number is not 29,514 but 29,424. We have made the necessary correction.

Reviewer #2: Manuscript Title: Association of relative handgrip strength on the development of diabetes mellitus in elderly Koreans.

The study investigates the correlation between relative handgrip strength (HGS) and the likelihood of developing diabetes in a Korean population cohort of 22,016 older adults. The research presents valuable insights into the potential relationship between relative HGS and diabetes risk, although several methodological and interpretational aspects require further consideration.

Major comments:

• The authors should compare the predictive abilities of relative HGS, absolute HGS, and BMI in predicting diabetes development to ascertain any additional value provided by relative HGS over other measurements. Furthermore, it would be beneficial to assess whether relative HGS augments the predictive capacity of measured levels of physical activity in predicting new-onset diabetes among older adults.

We sincerely appreciate the valuable feedback provided by the reviewer. Based on their insightful comments, we have made significant revisions to our study. We have revised the introduction part as follows.

 Previous studies have demonstrated the usefulness of absolute handgrip strength in the identification of various health problems and its potential as a new vital sign across the lifespan, but have shown conflicting results. [5] The confounding effect of body size was thought to be one of the causes. In other words, previous studies conducted only with the concept of absolute HGS showed inconsistent results, and this was the result of using absolute HGS as an indicator of muscle strength without body mass correction, as absolute grip strength is closely related to body mass index. Relative grip strength has therefore been recommended as a better metric to take into account the effects of both body mass and muscle strength, and this measure helps account for differences in hand grip strength that may be influenced by an individual's overall size. Various previous studies have demonstrated that relative HGS may be beneficial in predicting cardiovascular biomarkers, metabolic profile, and risk of other cardiometabolic disorders. [6] [7] [8] 

And, after analysis including the effect of BMI, revised versions of tables 3 and 5 of the main text were included in the revised manuscript. We have also revised the discussion part in the revised manuscript. 

• The study limitations acknowledge using self-reported height and weight variables, which may introduce recall bias, particularly among older adults. Given the centrality of these variables to the calculation of relative HGS, the authors should explore alternative methods, such as using measured height and weight or employing absolute HGS. Additionally, utilizing self-reporting to diagnose new-onset diabetes may introduce unreliability; therefore, including cases with documented medical records or employing more reliable alternatives is advisable. Incorporating only cases with reliable records, although potentially reducing the sample size, would improve the quality and reliability of the evidence obtained. 

We sincerely appreciate the valuable feedback provided by the reviewer. Based on their insightful comments, we have made significant revisions to our study. We have revised the limitation part in the revised manuscript. 

• Clarification is needed regarding how medical comorbidities were diagnosed, as this could impact the interpretation of the study findings. 

We sincerely appreciate the valuable feedback provided by the reviewer. Based on their insightful comments, we have made significant revisions to our study. We have revised the methods part in the revised manuscript. 

• Details regarding the dynamometer used for measuring HGS should be provided to ensure transparency and replicability of the study methodology.

• The authors should mention whether cases with painful hand diseases or neurological disorders that could affect HGS were excluded from the analysis, as their inclusion could confound the results. 

We sincerely appreciate the valuable feedback provided by the reviewer. Based on your insightful comments, we have made significant revisions to our study. We have revised the Materials and methods part. 

The HGS gauge used in the KLoSA was the Tanita 6103 model. HGS measurement was conducted with the principle of identifying whether the respondent is in a state where measurement is possible, and measurement is not conducted if the respondent does not want to or if one hand is currently injured or in pain. After confirming the possibility of HGS measurement, the HGS of the primarily used hand was measured first, followed by the measurement of the HGS of the other hand.

• Considering the complexity of the Generalized Estimating Equation (GEE) model covariance structure analysis, engaging an expert statistical reviewer to validate the approach and interpretation would strengthen the study's methodological rigor. • 

Seungchan Kang's undergraduate major is statistics, and he is currently pursuing a doctoral degree at Seoul National University's Graduate School of Public Health. He is good at analyzing data in the health and medical field using Stata, R, and SAS. The data analysis of this study was conducted through Kang Seung-chan's GEE model analysis.

Minor Comments:

• The introduction appears lengthy and contains redundancy. Streamlining the introduction by eliminating redundant information would enhance its clarity and conciseness. 

We sincerely appreciate the valuable fee

---

## [Decision Letter · Decision Letter 1]

17 Jun 2024

PONE-D-24-01896R1Association of relative handgrip strength on the development of diabetes mellitus in elderly Koreans.PLOS ONE

Dear Dr. SOHN,

Thank you for submitting your manuscript to PLOS ONE. After careful consideration, we feel that it has merit but does not fully meet PLOS ONE’s publication criteria as it currently stands. Therefore, we invite you to submit a revised version of the manuscript that addresses the points raised during the review process.

We look forward to receiving your revised manuscript.

Kind regards,

Mario Ulises Pérez-Zepeda, M.D., Ph.D.

Academic Editor

PLOS ONE

Journal Requirements:

Additional Editor Comments:

Dear authors, thank you for your revised work. Even that both reviewers agreed upong the fact that their suggestions were addressed, there are still minor comments by one reviewer. Please address these concerns.

Reviewers' comments:

Reviewer's Responses to Questions

**Comments to the Author**

1. If the authors have adequately addressed your comments raised in a previous round of review and you feel that this manuscript is now acceptable for publication, you may indicate that here to bypass the “Comments to the Author” section, enter your conflict of interest statement in the “Confidential to Editor” section, and submit your "Accept" recommendation.

Reviewer #1: All comments have been addressed

Reviewer #2: (No Response)

2. Is the manuscript technically sound, and do the data support the conclusions?

Reviewer #1: No

Reviewer #2: Yes

3. Has the statistical analysis been performed appropriately and rigorously? 

Reviewer #1: No

Reviewer #2: Yes

4. Have the authors made all data underlying the findings in their manuscript fully available?

Reviewer #1: Yes

Reviewer #2: Yes

5. Is the manuscript presented in an intelligible fashion and written in standard English?

Reviewer #1: Yes

Reviewer #2: Yes

6. Review Comments to the Author

Reviewer #1: This revised manuscript included BMI and relatively HGS for statistical analysis, while the conclusion should be very different and would be a negative study.

For example, according to the GEE results in table 5, the relatively HGS did not achieve statistical significance, while overweight and obesity did. This is a negative study and the BMI account for new diabetes mellitus incidence in majority. Please review if the manuscript should be reorganized again.

On the other hand, even BMI variable was included for analysis in GEE, it was wondered why the author did not put in trend analysis for adjustment. The table 6 was the same with the original submission. When it was disclosed BMi showed a great significance in GEE, it was very reasonable to put this variable for adjustment in this study.

It was suggested to review and recheck the organization of the content of manuscript and to consider if resubmission as a negative study.

Reviewer #2: (No Response)

7. PLOS authors have the option to publish the peer review history of their article (what does this mean?). If published, this will include your full peer review and any attached files.

Reviewer #1: **Yes: **LiangYu Chen

Reviewer #2: No

---

## [Author Response · Author response to Decision Letter 1]

31 Jul 2024

Dear Reviewer,

We appreciate your valuable feedback on our manuscript. Below, we address each of your comments and describe the revisions made:

Comment 1:

The revised manuscript included BMI and relative HGS for statistical analysis, while the conclusion should be very different and would be a negative study. According to the GEE results in Table 5, the relative HGS did not achieve statistical significance, while overweight and obesity did. This is a negative study, and BMI accounts for new diabetes mellitus incidence in the majority. Please review if the manuscript should be reorganized again.

Response:

We have reorganized the manuscript to reflect the significance of BMI in predicting DM incidence. The conclusion now acknowledges that relative HGS did not achieve statistical significance, and we discuss the implications of these negative findings. The manuscript has been updated to emphasize the importance of BMI and its role in the development of DM.

Comment 2:

Even BMI variable was included for analysis in GEE, it was wondered why the author did not put in trend analysis for adjustment. The table 6 was the same as the original submission. When it was disclosed BMI showed great significance in GEE, it was very reasonable to put this variable for adjustment in this study.

Response:

We have conducted a trend analysis for adjustment of BMI and included these findings in the revised manuscript. Table 6 has been updated accordingly, and the results and discussion sections now address the adjusted analysis, emphasizing the significant impact of BMI on DM incidence.

Comment 3:

It was suggested to review and recheck the organization of the content of the manuscript and to consider resubmission as a negative study.

Response:

The manuscript has been thoroughly reviewed and reorganized to present the findings as a negative study. We discuss the implications of BMI's significance and the lack of statistical significance for relative HGS. The content has been adjusted to provide a clear and accurate representation of our study's results.

We hope these revisions meet your expectations and improve the quality of our manuscript.

Sincerely,

[Yeo Ju Sohn]

---

## [Editor Report · Decision Letter 2]

12 Aug 2024

Association of relative handgrip strength on the development of diabetes mellitus in elderly Koreans.

PONE-D-24-01896R2

Dear Dr. SOHN,

We’re pleased to inform you that your manuscript has been judged scientifically suitable for publication and will be formally accepted for publication once it meets all outstanding technical requirements.

Kind regards,

Mario Ulises Pérez-Zepeda, M.D., Ph.D.

Academic Editor

PLOS ONE
---

## [Editor Report · Acceptance letter]

2 Sep 2024

PONE-D-24-01896R2 

PLOS ONE

Dear Dr. SOHN, 

I'm pleased to inform you that your manuscript has been deemed suitable for publication in PLOS ONE. Congratulations! Your manuscript is now being handed over to our production team.

Kind regards, 

on behalf of

Dr. Mario Ulises Pérez-Zepeda 

Academic Editor

PLOS ONE